# Evaluation of Accuracy and Performance of a Novel, Fully Gantry Integrated 3D Laser System for Computed Tomography Guided Needle Placement: A Phantom Study

**DOI:** 10.3390/diagnostics13020282

**Published:** 2023-01-12

**Authors:** Katharina Weigel, Rahel A. Kubik-Huch, Matthias Baer-Beck, Christian Canstein, Michael Kostrzewa

**Affiliations:** 1Department of Radiology, Baden Cantonal Hospital, 5404 Baden, Switzerland; 2Barking, Havering and Redbridge University Hospitals NHS Trust, Romford RM7 0A, UK; 3Siemens Healthineers, 91301 Forchheim, Germany; 4Institute of Clinical Radiology and Nuclear Medicine, University Hospital Mannheim, Heidelberg University, 68167 Mannheim, Germany

**Keywords:** computed tomography, intervention, laser-guidance, phantom, safety, radiation exposure

## Abstract

The purpose of this phantom study was to compare the accuracy, speed and technical performance of CT guided needle placement using a conventional technique versus a novel, gantry integrated laser guidance system for both an expert and a novice. A total of 80 needle placements were performed in an abdominal phantom using conventional CT guidance and a laser guidance system. Analysis of pooled results of expert and novice showed a significant reduction of time (277 vs. 204 s, *p* = 0.001) and of the number of needle corrections (3.28 vs. 1.58, *p* < 0.001) required when using laser guidance versus conventional technique. No significant improvement in absolute (3.81 vs. 3.41 mm, *p* = 0.213) or angular deviation (2.85 vs. 2.18°, *p* = 0.079) was found. With either approach, the expert was significantly faster (conventional guidance: 207 s vs. 346 s, *p* < 0.001; laser guidance: 144 s vs. 264 s, *p* < 0.001) and required fewer needle corrections (conventional guidance: 4 vs. 3, *p* = 0.027; laser guidance: 2 vs. 1, *p* = 0.001) than the novice. The laser guidance system helped both the expert and the novice to perform CT guided interventions in a phantom faster and with fewer needle corrections compared to the conventional technique, while achieving similar accuracy.

## 1. Introduction

Minimally invasive, image guided procedures are a mainstay of modern medicine. A broad spectrum of minimally invasive procedures are performed using ultrasound (US) and fluoroscopic or computed tomography (CT) guidance, ranging from tissue biopsies and abscess drainages to the more complex procedures of interventional oncology [1].

For difficult to reach, deep lying targets, and especially for percutaneous minimally invasive tumor ablation, precise and reproducible CT guided needle placement is paramount. Yet, especially double-angled, out-of-plane trajectories have traditionally been difficult to realize with conventional CT guidance [2]. In a phantom study, Kostrzewa et al. could show that out-of-plane trajectories using conventional CT guidance take longer, i.e., are more difficult to realize, than in-plane trajectories [3]. To circumvent this limitation, CT manufacturers have developed dedicated software solutions for needle guidance. However, these purely software based solutions do not directly support the interventionalist in needle placement in the CT suite, but mostly aid in planning the needle trajectory.

To mitigate this, multiple third-party solutions have been developed [4,5,6,7]. Whereas the technical approach of these solutions differs, all of them require additional, external hardware and/or software and need to be set up separately from the CT scanner, thus constituting additional time expenditure and cost.

Recently, a new CT scanner equipped with a full gantry integrated system for needle guidance entered the market (Somatom X.ceed, Siemens Healthineers, Forchheim, Germany). The system is comprised of four gantry mounted fan beam lasers and a dedicated software solution.

The aim of this phantom study was to compare the accuracy, speed and technical performance of CT guided needle placement using the conventional technique versus a workflow utilizing the novel, gantry integrated 3D laser guidance system for both an expert and a novice user. Our hypothesis is that the 3D laser guidance system allows for faster needle placement with fewer needle corrections compared to conventional CT guidance.

## 2. Materials and Methods

Given that the performed study was a phantom study, no institutional review board approval was required.

### 2.1. Technical Functionality of the Novel 3D Laser MDCT Guidance System

The novel 3D laser guidance system (myNeedle Companion, Siemens Healthineers, Germany) is comprised of four dedicated laser projectors, mounted at nine, eleven, one, and three o’clock positions on the housing of the scanner gantry (Somatom X.ceed, Siemens Healthineers, Germany, Figure 1). A dedicated software solution (my Needle Guide, Siemens Healthineers, Germany) is an integral component of the system.

Each of the projectors is composed of a rotatable fan laser and a rotatable mirror placed opposite ththe fan laser (Figure 2). By adjusting the mirror angle and fan laser angle, the orientation and position of the fan laser can be adjusted within the operating volume (Figure 2 and Figure 3). By adjusting the laser rotation angle and mirror rotation angle, the system can match the intersection line of the two fan lasers to the planned needle path. To optimize accuracy, the system selects two out of the four laser projectors which are best suited to visualize the geometry of the planned needle path. The fan lasers are classified as Class 2 lasers based on their energy and their danger level, and thus no special safety measures such as wearing safety glasses are required.

The workflow of 3D laser guided interventions is comparable to that of conventional CT guided needle placement. After volume acquisition, a needle trajectory is planned in a dedicated software solution (myNeedle Guide, Siemens Healthineers, Forchheim, Germany). Multiple in-plane or double-angulated (out-of-plane) needle paths can be planned at the same time. After planning, the user can select a path for visualization with the laser guidance system. The table moves automatically out of the gantry, stopping at an optimal visualization position. Two projectors project the planned trajectory onto the operating volume. Whereas the laser crosshair projected onto the surface constitutes the needle insertion point, the angulation of the intersection line of the two fan lasers constitutes the angulation of the needle trajectory. Accordingly, the needle is inserted into the volume at the laser crosshair and aligned along the intersection line of the fan lasers (Figure 4). Insertion depth is controlled manually.

### 2.2. Definition of Needle Trajectories

In a multi-modality abdominal phantom (model 057A, CIRS, Norfolk, VA, USA), eight targets were identified (six liver lesions, two renal pelvises). In total, ten different in-plane and ten different out-of-plane needle trajectories were planned for these lesions. In-plane trajectories constitute trajectories angulated only in the transverse plane of the CT scanner, whereas out-of-plane trajectories are also angulated in the coronal and sagittal planes. Using a 15 cm 21 G graduated Chiba needle (Cook Medical, Bloomington, Indiana, USA), each lesion was punctured along the previously defined trajectories using a conventional needle guidance (CNG) technique and by using the novel gantry integrated 3D laser guidance (3DLG) system. Each intervention was performed once by an expert user (more than 10 years of experience in interventional radiology) and once by a novice (resident, no experience in radiology or interventional radiology), resulting in a total of 80 needle placements. The novice performed one training needle placement each with the conventional and the laser guidance in order to familiarize herself with the system’s functionality. All interventions were performed over the course of three days, and then first all in-plane and then all out-of-plane interventions were performed. Needle placements were first performed using CNG for all 20 trajectories, then using 3DLG. The endpoint of each intervention was hitting the target lesion with the needle tip as centrally as possible.

### 2.3. Scan Parameters

Conventional guided interventions and 3D laser guided interventions were performed on the same MDCT scanner (Somatom X.ceed, Siemens Healthineers, Germany) and scan parameters were identical. Settings for spiral CT were: 173 slices at 1.0 mm slice thickness, 100 kV tube voltage with tin filter, effective tube current of 256 mAs. Settings for sequential CT were: 23 slices at 1.0 mm slice thickness, 120 kV tube voltage with tin filter, effective tube current of 116 mAs. 

### 2.4. Conventional Needle Guidance (CNG)

After CT volume acquisition of the phantom, each needle trajectory was planned as initially defined. The needle insertion point was measured from the center line and was transferred onto the phantom using the positioning lasers and a tape measure. Using sequential CT guidance, the Chiba needle was introduced into the phantom, estimating the insertion angle. The needle path was corrected under sequential CT view until the endpoint was met (i.e., the lesion was hit).

### 2.5. D Laser Guidance (3DLG)

Phantom interventions using the 3D laser guidance system were performed analogue to conventional CT guidance. After initial volume acquisition of the phantom, each needle trajectory was planned as initially defined. Laser projection was then started and the needle was inserted into the phantom as described above. Needle position was controlled and corrected using sequential CT scans until the endpoint was met (i.e., the lesion was hit).

### 2.6. Data Collection and Statistical Analysis

Overall procedural times, defined as the time from the start of each intervention (for conventional guidance: start of entry point measurement; for laser guidance: start of laser projection) until the final needle position was reached were recorded. Numbers of needle corrections until the final needle position was obtained were counted. An evaluation of final needle position relative to planned needle trajectory and target lesion was performed on the last sequential CT scan using multi planar reconstruction. As published by Kostrzewa et al., using multi-planar reconstruction angular deviation between planned path and needle position, as well as absolute deviation—defined as distance from needle tip to center of the lesion—were measured (compare Figure 2 in [3]). Primary outcomes compared were overall results of conventional versus laser guidance. A subgroup analysis was undertaken to compare in- versus out-of-plane procedures and the results of expert versus novice when using either guidance system.

An analysis of all data outcomes was performed using IBM SPSS Statistics (version 28.0.0.0). All variables underwent a related samples Wilcoxon Signed rank test as not all data sets were normally distributed. Results with a two tailed *p*-value lower than 0.05 were considered statistically significant (95% confidence interval). All graphics were created using Microsoft Excel (version 16.55).

## 3. Results

All interventions were technically successful meaning that the respective, targeted lesion was hit in all 80 interventions. Trajectory length ranged from 48 to 95 mm (mean 77.65 mm, IQR 72.75–88.5) and lesion diameter ranged from 8.2 to 13.7 mm (mean 10.8 mm).

In a pooled data analysis of expert and novice interventions with conventional needle guidance (CNG), absolute deviation from needle tip to the center of the lesion ranged from 0 to 7.07 mm (mean 3.81 ± 1.74 mm) and angular deviation ranged from 0 to 8° (mean 2.85 ± 1.87°). Mean number of needle insertions until final placement was 3.28 (range 2–6) and mean overall procedural duration was 276.73 s (±132.92 s).

For pooled data of expert and novice interventions with 3D laser guidance (3DLG), the absolute deviation from needle tip to center of the lesion ranged from 1.39 to 5.21 mm (mean 3.41 ± 1.14 mm) and the angular deviation ranged from 0 to 5° (mean 2.18 ± 1.43°). The mean number of needle corrections until final placement was 1.58 (range 1–3) and the mean overall procedural duration was 204 s (±103.15 s).

When comparing CNG with 3DLG, a significant reduction of procedural duration (*p* = 0.001) and of number of needle corrections (*p* < 0.001) could be found when using the laser guidance system (Figure 5). This was true for the novice and the expert alike. When analyzing pooled data of novice and expert in-plane and out-of-plane procedures, a significant reduction in procedural time when using 3DLG could only be found for out-of-plane procedures (308 s vs. 192 s, *p* = 0.001).

The novice showed a reduction in procedural duration from 346 s to 264 s (*p* = 0.013) and a reduction of needle corrections from 3.55 to 2 (*p* < 0.001), whereas the expert showed a reduction in time from 207 s to 144 s (*p* = 0.017) when using 3DLG. The expert was found to be significantly quicker in needle placement than the novice when using both CNG (207 s vs. 346 s, *p* < 0.001) and 3DLG (144 s vs. 264 s, *p* < 0.001). The only exception was out-of-plane procedures with CNG (261 s vs. 356 s, *p* = 0.103).

Overall, the novice required significantly more needle corrections than the expert, averaging at 4.00 under CNG and 2.00 under 3DLG versus the expert at 3.00 and 1.00 (*p* = 0.027 for CNG, *p* < 0.001 for 3DLG), respectively. Novice and expert alike showed a reduction of needle corrections from 3.55 to 2.00 (*p* < 0.001) and 3.00 to 1.15 (*p* < 0.001), respectively, when using 3DLG (Figure 6).

In the pooled data of the expert and novice, the absolute deviation decreased from a mean of 3.81 ± 1.74 mm using CNG to 3.41 ± 1.14 mm (*p* = 0.113) using 3DLG, and angular deviation decreased from 2.85 ± 1.87° to 2.18 ± 1.43° (*p* = 0.109), respectively (Figure 7 and Figure 8). The novice showed a significant decrease of absolute deviation from 4.21 ± 1.73 to 3.44 ± 1.11 mm when using laser guidance (*p* = 0.037). The expert was not significantly more accurate than the novice, neither with CNG nor with 3DLG (Figure 8).

## 4. Discussion

Currently, an increasing number of second party vendor systems for CT needle navigation are becoming available. The driving force of this development is the steadily increasing importance of image guided, needle-based procedures in clinical routine [1,8]. Besides standard procedures such as biopsies and abscess drainages, procedures of interventional oncology, such as minimally invasive, percutaneous tumor ablation, can especially profit from these systems. To date, several publications have demonstrated the value of guidance systems in interventional oncology. Bale et al. showed that, by using a navigation system for needle guidance, liver lesions, such as hepatocellular carcinomas and cholangiocarcinoma cells, as well as colorectal liver metastasis, can safely and effectively be treated by percutaneous radiofrequency ablation irrespective of the size of the lesion [9,10,11]. Schaible et al. showed that, by using a navigation system, the primary efficacy of microwave ablation of liver lesions was significantly greater than when using conventional CT guidance alone [7].

An integrated cone beam computed tomography (CBCT) solution for needle guidance using laser projection was initially presented in 2014 [3]. While a sufficient accuracy of the system as well as feasibility and utility of CBCT guided percutaneous interventions could be shown [3,12,13], clinical real-world usage remains infrequent and is restricted to specialized centers and applications. Reasons include, amongst others, insufficient soft tissue contrast of CBCT, the complexity of conducting CBCT guided procedures due to large systems that freely rotate around the patient, and the availability of these systems in clinical day to day routines. By adapting the principle of a laser guidance system to a CT scanner, these limitations of CBCT are overcome.

In our phantom study, we showed that a novice user and an expert both profit from the laser guidance system, resulting in an improvement of procedural times and needle corrections required until a defined target was reached for both. Whereas the expert was faster than the novice and required fewer needle corrections with both approaches, no difference in accuracy of needle placement between novice and expert could be found. By using 3DLG, the novice was able to achieve the same accuracies as the expert faster and with fewer needle corrections when compared with CNG, showing the potential value of the system, especially for less experienced users.

Whereas the average deviation of 3.41 ± 1.14 mm for 3DLG is acceptable, it is higher than deviations reported in the literature for other guidance systems [4,6,14,15]. This may be due to the design of the system, which only uses optical guidance but no physical guiding tool such as a needle holder, which is the case for most other comparable guidance systems. This may therefore constitute a future area of improvement of the system.

In contrast to other guidance systems, such as Cascination, the xAct, Perfint or Zerobot robot, Imactis and a recently published system by Fong et al., a key advantage of the presented system is its full integration into a diagnostic CT scanner [4,6,14,15,16,17,18]. This omits the need for additional, external hardware and/or software, which needs to be set up separately from the CT scanner requiring additional time and resulting in higher cost. In a paper by Levy et al., procedural times from 30 to 45 min are described for robot guided percutaneous biopsies. This is probably still too long to reach routine clinical acceptance [4]. Average set up time for the electromagnetic navigation system described in a paper by Ringe et al. was 6.75 min [14].

A key finding was the significant reduction of needle corrections required until the lesion was hit when using 3DLG in comparison to CNG for both the expert and the novice. While this may result in fewer complications, complications of CT guided interventions are already reported as being very low [19]. However, a lower number of needle corrections will most likely result in lower overall radiation exposure in real clinical scenarios. In a paper by Rathmann et al., radiation exposure for the interventionalist was shown to be generally low for CT guided procedures, but with higher radiation exposure for the interventionalist in more complex interventions [2]. Since fewer needle corrections result in fewer control scans, radiation exposure for both patient and interventionalist are likely to be lower when using laser guidance in clinical routine. This remains a field to be investigated and a study with this focus is currently being undertaken at our institute.

A potential limitation of this study is the definition of the endpoint of each intervention being “hit lesion as centrally as possible”. This is most likely what has led to no difference in accuracy when using the 3DLG system versus CNG, thus skewing the results towards number of needle corrections and overall procedural time. A further limitation of this study is the fact that a phantom study was performed. Hence, relevant factors such as patient movement and breathing motion are not considered when analyzing the time efficiency, accuracy, and safety of the system. However, one may argue that, by eliminating these factors, a better comparability between conventional and laser guidance could be achieved. Especially since the same, predefined trajectories could be used for both systems, it is unlikely that lesion size, shape, location and trajectory length had an effect on our findings. Further, given that the same lesion was punctured multiple times, one may argue that a previous needle placement affected the next needle placement. By first performing all CNG and then all 3DLG interventions and using a thin needle, which hardly leaves a mark in the phantom, we tried to minimize this effect. Lastly, the fact that only two participants performed the interventions limits the validity of the study with respect to the effect of 3DLG on different participants. However, the main goal of the study was to compare 3DLG to CNG.

## 5. Conclusions

In this phantom study we showed that using a novel 3D laser guidance system integrated into a CT gantry resulted in significantly faster needle placement with fewer needle corrections than with conventional MDCT guidance. This was the case for both a novice user and an expert, while resulting in a similar accuracy of needle placement. Decreased procedure times may allow for increased case load. Fewer needle corrections may reduce the risk of procedural complications and will most likely result in less radiation exposure for patient and interventionalist in clinical routine. The fact that the novice achieved similar accuracies in needle placement without any previous experience in CT guided interventions demonstrates the intuitive design of the system. The 3D laser guidance system may thus help less experienced interventionalists to conduct CT guided procedures efficiently and safely. The full integration of the system into the CT gantry omits set up times and makes the usage of the system efficient for clinical routine.

## Figures and Tables

**Figure 1 diagnostics-13-00282-f001:**
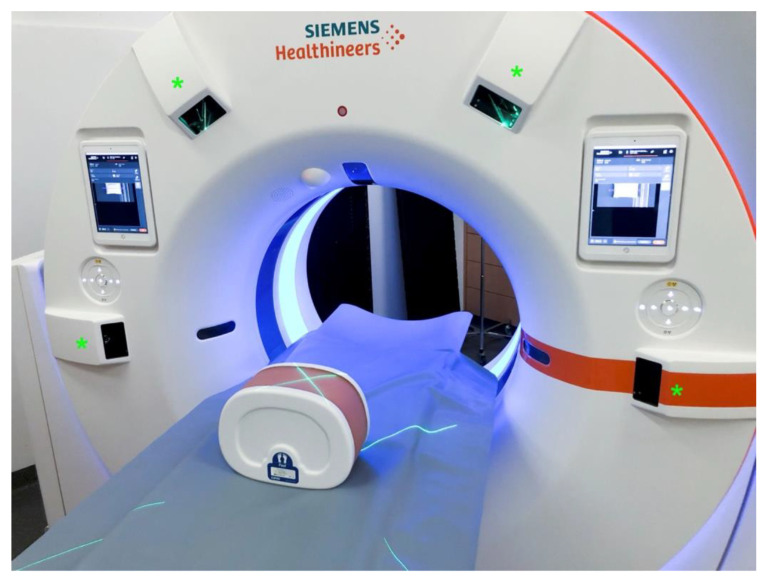
Gantry with four laser projectors mounted onto the housing (green asterisk). Visualization of a planned needle path by two intersecting fan beam lasers projected onto the phantom.

**Figure 2 diagnostics-13-00282-f002:**
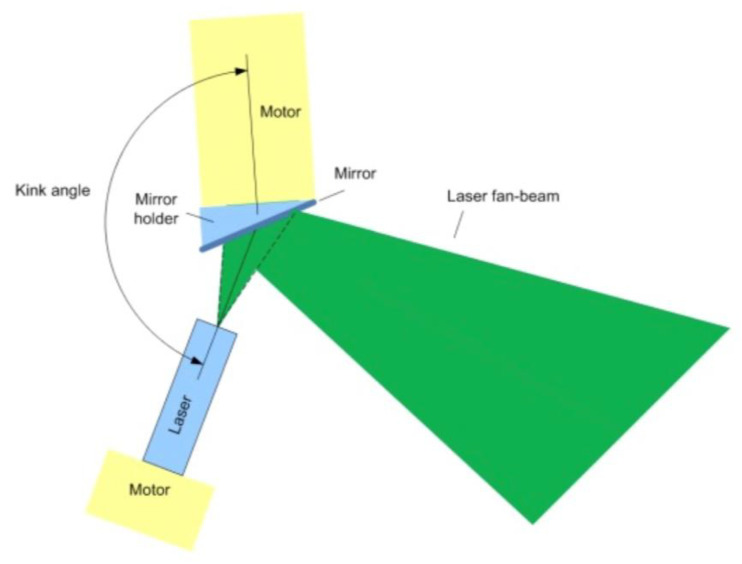
Schematic of a projector: each of the four projectors is comprised of a laser and a mirror. The orientation of the fan laser and the mirror can be adjusted by a rotation motor.

**Figure 3 diagnostics-13-00282-f003:**
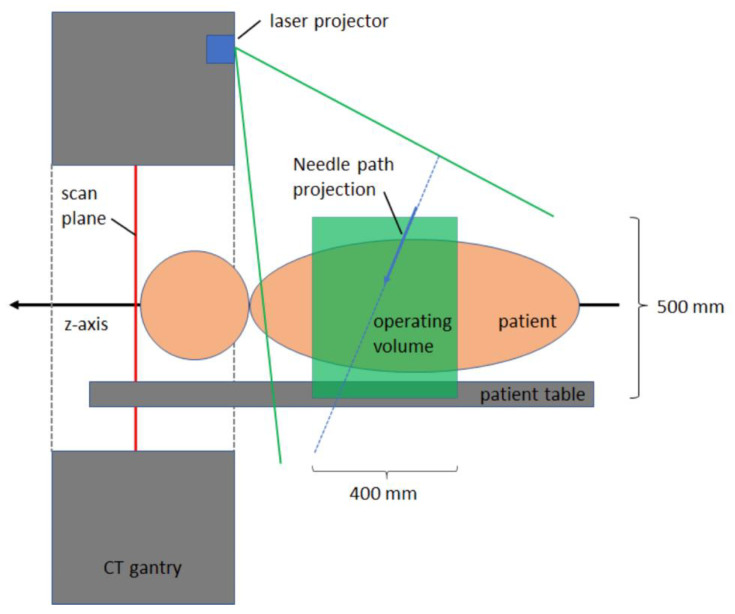
Schematic of needle path projection in front of the gantry onto the operating volume. Within the operating volume, a visualization of needle paths is possible.

**Figure 4 diagnostics-13-00282-f004:**
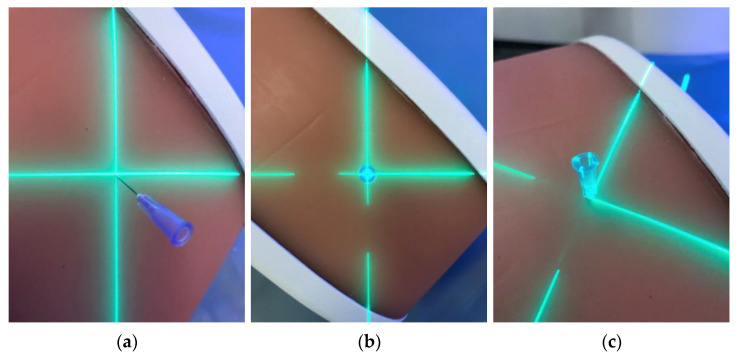
Fan beam laser projection onto the phantom and respective alignment of the needle (**a**–**c**). Precise alignment of the needle along the laser is paramount to achieve accurate needle placement.

**Figure 5 diagnostics-13-00282-f005:**
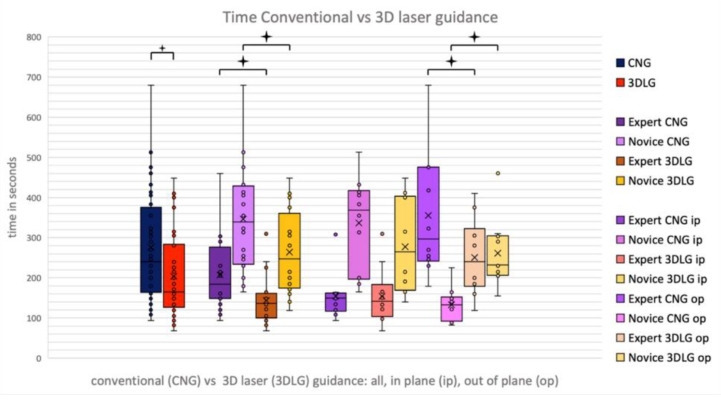
Procedural times for conventional needle guidance (CNG) and 3D laser guidance (3DLG), for expert and novice in in-plane and out-of-plane procedures.

**Figure 6 diagnostics-13-00282-f006:**
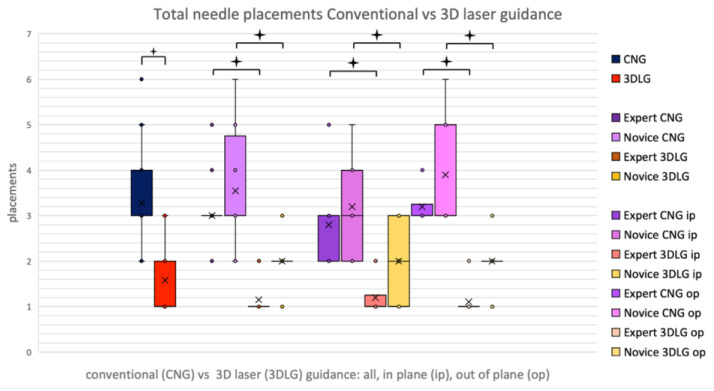
Total needle corrections for conventional needle guidance (CNG) and 3D laser guidance (3DLG), for expert and novice, in in-plane and out-of-plane procedures.

**Figure 7 diagnostics-13-00282-f007:**
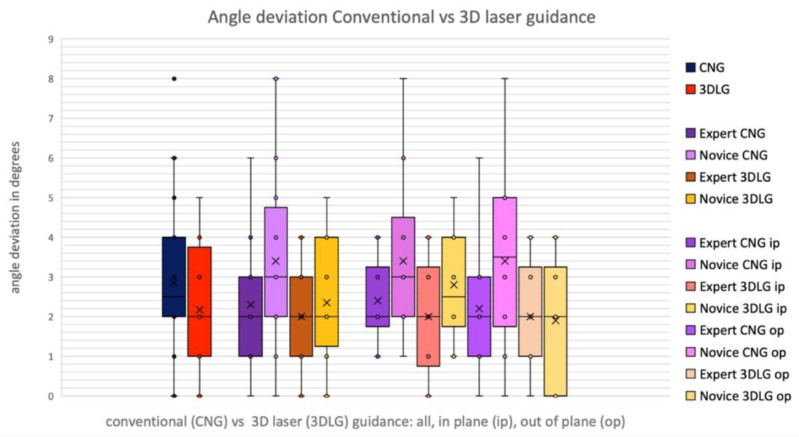
Angular deviation for conventional needle guidance (CNG) and 3D laser guidance (3DLG), for expert and novice, in in-plane and out-of-plane procedures.

**Figure 8 diagnostics-13-00282-f008:**
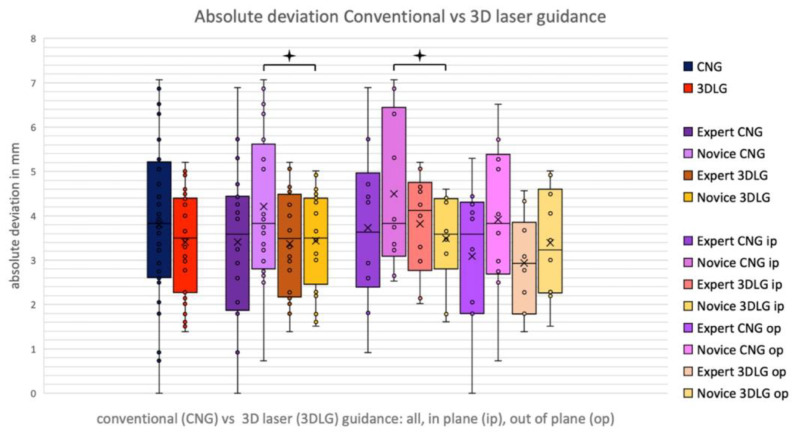
Absolute deviation for conventional needle guidance (CNG) and 3D laser guidance (3DLG), for expert and novice in in-plane and out-of-plane interventions.

## Data Availability

Data sharing is not applicable to this article.

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
