# Peer review of "Evaluation of Accuracy and Performance of a Novel, Fully Gantry Integrated 3D Laser System for Computed Tomography Guided Needle Placement: A Phantom Study"

_diagnostics, 2023, doi:10.3390/diagnostics13020282_

Round 1

Reviewer 1 Report

Dear authors

Your study is very interesting however the limitation of having it performed only on two subjects is a strong limitations.

I would suggest to perform the study including more participants and improve the introduction citing existing literature were needed.

Elements in need of attention are:

line 32: needs citation

line 36: needs citation

line 40: not clear why a "non direct" solution represent a problem, there should be more to an integrated hardware than time and cost.

line 53: state an hypothesis

line 120: the users are repeating tasks on the same phantom therefore is not possible discern the knowledge of the phantom from the assistance received by the laser system. Multiple subjects are needed.

line 155: detail how the angular deviation is measured.

Figures: the fonts are too small compared to the text

lines 241: 260 needs citation

line 216: no need to restate the purpose

line 301: the main limitation is the use of only 2 users that were repeating tasks. this need to be address

Reviewer 2 Report

The objective of this study is interesting. The manuscript was well-prepared. The statistical analysis is appropriate. However, there're some queries regarding the methodology of this study;

1. Please provide the reason why only one expert and one novice participated in this study. To recruit more participants would make the result more reliable

2. As there's only one participant in each group, it could not be concluded that both groups achieved the same accuracy in needle placement.

3. Please define the novice in this study (general practitioner or radiologist or resident or fellow)

4. Did both participants have any experience on this phantom?

Round 2

Reviewer 2 Report

The revised version was well-written. Very few participants might limit the generalizability. Further validity study is required before clinical application.